# New Challenges for Anatomists in the Era of Omics

**DOI:** 10.3390/diagnostics13182963

**Published:** 2023-09-15

**Authors:** Anna Maria Stabile, Alessandra Pistilli, Ruggirello Mariangela, Mario Rende, Desirée Bartolini, Gabriele Di Sante

**Affiliations:** 1Department of Medicine and Surgery, Section of Human, Clinical and Forensic Anatomy, University of Perugia, 60132 Perugia, Italy; anna.stabile@unipg.it (A.M.S.); alessandra.pistilli@unipg.it (A.P.); mariangela11@live.it (R.M.); mario.rende@unipg.it (M.R.); 2Department of Pharmaceutical Sciences, University of Perugia, 06126 Perugia, Italy

**Keywords:** genomics, transcriptomics and proteomics, spatial profiling and pathomics, connectomics and neural networks, 3D bioprinting and artificial intelligence

## Abstract

Anatomic studies have traditionally relied on macroscopic, microscopic, and histological techniques to investigate the structure of tissues and organs. Anatomic studies are essential in many fields, including medicine, biology, and veterinary science. Advances in technology, such as imaging techniques and molecular biology, continue to provide new insights into the anatomy of living organisms. Therefore, anatomy remains an active and important area in the scientific field. The consolidation in recent years of some omics technologies such as genomics, transcriptomics, proteomics, and metabolomics allows for a more complete and detailed understanding of the structure and function of cells, tissues, and organs. These have been joined more recently by “omics” such as radiomics, pathomics, and connectomics, supported by computer-assisted technologies such as neural networks, 3D bioprinting, and artificial intelligence. All these new tools, although some are still in the early stages of development, have the potential to strongly contribute to the macroscopic and microscopic characterization in medicine. For anatomists, it is time to hitch a ride and get on board omics technologies to sail to new frontiers and to explore novel scenarios in anatomy.

## 1. Introduction

Anatomy, throughout the history of medicine, has overcome different challenges, contributing to human progress. Even in ancient civilizations, such as ancient Egypt and Greece, basic anatomical knowledge was acquired through the dissections of animals and limited human cadavers [1]. Tools like knives and scalpels were used for these dissections. Andreas Vesalius, considered the father of anatomic studies, produced “De humani corporis fabrica” (On the Fabric of the Human Body) in 1543 [2]. This work, aided by the printing press, contained detailed illustrations of the human body, helping to disseminate accurate anatomical information widely. Anatomical theatres emerged in the 16th century, providing venues for public dissections and medical education. The production of anatomical models from wax and other materials also helped educators teach anatomy more effectively [3]. The invention of the microscope in the 17th century allowed anatomists to explore the infinite complexity of cells and tissues [4]. This led to the discovery of previously unseen structures and paved the way for advancements in histology (the study of tissues).

Since the 18th century, Europe has been the center of modern anatomy, with the rapid growth of different medical schools in England [5,6] and France [7]. In Italy, just to name a few, we had the examples of the Anatomical School of Padua [8] and of Morgagni [9], considered the father of modern anatomical pathology, and, after him, the field of anatomy continued to evolve. In line with this, since the invention of optical microscopy, numerous advances have led to a deeper understanding of anatomy at the cellular level, paving the way for new discoveries in physiology.

The 19th century saw the establishment of medical schools throughout Europe and North America, which provided a formalized system for training physicians and medical researchers. Anatomical dissection became a critical component of medical education [10].

This paved the way for new perspectives and new approaches in anatomical studies, with the introduction of imaging techniques in the 20th century, when advancements were made in technology such as X-rays, MRI, and CT scans [11]. More recently, the advent of computer-assisted imaging and machine learning [12] allowed for the non-invasive visualization of internal anatomy, which has changed medical diagnosis and treatment [13].

For instance, in the 100 years following Golgi’s death, many advances have also been made in microscopic anatomy [14]. The best example is represented by the field of neuroscience, where, in recent decades, new milestones have been set, such as the identification of novel neurotransmitters and receptors [15,16,17], and the development of new drugs to treat neurological disorders. All these discoveries have also been made possible following the mapping of the human genome, which has led to new insights into the genetic basis of neurological diseases, and the development of new imaging techniques, such as magnetic resonance imaging (MRI) and computed tomography (CT), which have revolutionized our ability to study the structure and function of the brain [18].

Anatomical studies have faced several challenges in recent years, including several limitations; for instance, access to cadavers is strongly reduced when compared with the previous century, and, although dissections are essential for the study of human anatomy, obtaining cadavers can be difficult due to ethical and legal considerations, with a big range of differences among distinct countries: for example, the concept of body donation is culturally unthinkable in certain countries [19]. This can limit the availability of study materials for anatomists. In recent years, we assisted an expansion towards “a good body donation practice” in countries with resistant cultural/religious backgrounds, with a strong debate in terms of bioethics, and with an evolution of the regulation and methods/limits in south Europe [20,21] and worldwide [22,23,24].

Technological advances have the potential to transform the way cadaveric preservation and anatomical studies are approached and the impact they have on medical education, training, and research. However, it is essential to balance these innovations with ethical considerations and a thoughtful approach to ensure the best outcomes for medical professionals and patients alike. As technology advances, discussions about the ethical use of cadavers and alternative preservation methods become more important. Ensuring that these advancements are used responsibly and respectfully is a crucial aspect of their development.

Furthermore, the decrease in interest in anatomy is due both to the rise of modern technology in other scientific fields and to anatomical diversity because of globalization (historically, cadavers used for anatomical studies were predominantly white males).

Finally, an issue, not only with regard to anatomical studies but science in general, is the access to technology: the availability of high-quality imaging and other technology can be limited in certain areas, hindering research and education.

Despite these challenges, anatomical studies continue to be a crucial part of medical and biological research, and new techniques and technologies are being developed to overcome these obstacles. In this regard, recently, anatomical studies confirmed the discovery of new insight on macroscopic and microscopic brain anatomy, demonstrating the presence of a mesothelium subdividing the subarachnoid space into functional compartments [25].

Overall, the field of anatomy has continued to evolve and expand in the centuries as new technologies and research methods have emerged. For these reasons, the recent advancements in omics technologies and novel imaging techniques should help anatomists by providing them with new tools for exploring the morphological features of biological specimens at various scales, from molecules to organs. In this manuscript, we will discuss the use of omics techniques and novel technologies in anatomic studies, their advantages, limitations, and the opportunities that can represent novel frontiers.

## 2. Discussion

### 2.1. Omics Techniques

The term “omics” refers to a set of high-throughput methods for the analysis of large datasets of biological molecules, such as DNA, RNA, proteins, and metabolites or even techniques generating and managing a high amount of data to dissect a specific aspect. Omics technologies, such as genomics, transcriptomics, proteomics, and metabolomics, enable the characterization of the molecular composition of tissues and organs, allowing for a comprehensive understanding of their structure and function, while other more recent 3D and computer-assisted “omics”, such as radiomics, pathomics, and connectomics, and new tools such as neural networks, 3D bioprinting, and artificial intelligence [26], have the potential to contribute to the definition of macroscopic features in medicine in general and in particular in anatomy (Figure 1 and Table 1).

### 2.2. Genomics

A good paradigm of omics impact in sciences is represented by genomics, probably the first and the most implemented field in recent years. This technique indicates the study of an organism’s whole DNA sequence, including genes and non-coding regions. This technique exploded after human genome sequencing [60,61] and, so far, allowed to identify genetic variations that may affect an organism’s development, function, and disease susceptibility [62]. In anatomic studies, genomics has been used to investigate the genetic basis of morphological traits and developmental processes [63,64,65,66]. In human, forensic and evolutionary anatomy, the usage of genomics brought several advantages such as (i) allowing for a more comprehensive understanding of the genetic basis of anatomical structures and functions; (ii) revealing previously unknown genetic variations that contribute to anatomical diversity and that can implement evolutionary studies; (iii) reconstructing the evolutionary relationships between different anatomical structures and tracing their evolution over time [67,68,69,70]. Recent advancements in genomics and molecular biology have provided insights into the genetic basis of anatomical variation and disease susceptibility. This deeper understanding has implications for personalized medicine and targeted therapies.

Advances in genomic technology have made it possible to analyze genetic data at a faster rate and with greater accuracy than ever before, being implemented in human anatomic studies, among others. However, the sheer amount of data generated by genomics studies can be overwhelming, and analysis and interpretation can be challenging [71,72,73,74]. In addition, in the era of data exchange and strict laws about privacy, there may be ethical concerns around the collection and use of genetic data, particularly when it comes to human subjects without informed consent [75,76,77].

Recent advances in genomics have led to the development of new techniques for analyzing and visualizing complex genetic data [61,62,78,79]. For example, researchers are using machine learning algorithms to identify patterns in genetic data that would be difficult for humans to discern. Additionally, there is an increasing interest in using genomics to study the relationship between organs/systems and diseases, which could lead to new treatments and therapies, in the perspective of the clinical anatomy [80,81,82]. Finally, as the cost of genomic sequencing continues to drop, genomics is becoming more accessible to researchers and clinicians around the world.

### 2.3. Transcriptomics and Epigenomics

Together with genomics the advent of transcriptomics, which analyzes the entire set of all types of RNA transcripts, and provides information on gene expression levels, alternative splicing, and post-transcriptional modifications, had an important impact in science. In tandem and in strict relation with both transcriptomics and genomics, epigenomics allowed for the study of how changes in gene expression occur due to modifications to DNA that do not involve changes to the underlying DNA sequence. These modifications can be influenced by environmental factors, lifestyle choices, and other external influences [83,84,85].

In anatomic studies, both transcriptomics and epigenomics can be used to explore gene expression patterns associated with tissue and organ development, differentiation, and function, showing how these modifications may contribute to the development and progression of diseases, and how they may be influenced by various factors, even environmental ones. For example, researchers may use epigenomics to study how changes in DNA methylation patterns may contribute to the development of cancer [13,86], or how changes in histone modifications may play a role in the development of neurological [87,88,89] or immune-mediated disorders [90,91,92].

Transcriptomics and epigenomics can provide a detailed picture of the genes and of their regulators that are active in a particular tissue or cell type. This can help to elucidate their functions, contributing to the identification of differences in gene expression between different cell types or tissues, and providing insight into their molecular and functional diversity [93,94].

Advances in transcriptomic and epigenomic technologies have made it possible to analyze gene expression data at a high throughput and with high resolution and the underlying mechanisms of homeostasis/disease, identifying potential targets for therapies and interventions. Moreover, the combination and the validation of multiple data available in the literature made possible the atlas of human tissues generation [34]. There are still many unanswered questions about how gene expression and epigenetic data relate to anatomical structures and functions, and this open field could represent a challenge and a future frontier of anatomy.

Recent advances in transcriptomics and epigenomics have led to the development of new techniques for analyzing and visualizing gene expression data. For example, single-cell RNA sequencing has made it possible to identify and characterize previously unknown cell types in complex tissues. Additionally, researchers are increasingly interested in using transcriptomics and epigenomics to study the relationship between gene expression and disease, which could lead to the identification of new therapeutic targets. Finally, as the cost of both omics continues to drop, it is becoming more accessible to researchers and clinicians around the world and transforming these technologies leads to powerful tools in anatomic studies that can help researchers to better understand the underlying mechanisms of disease and identify potential targets for therapies and interventions.

### 2.4. Proteomics, Metabolomics and Interactomics

Proteomics and Metabolomics provide data about the expression and post-translational modifications of proteins, their interactions, composition, and metabolite concentration. In anatomic studies, proteomics can be used to identify protein networks and metabolic pathways associated/involved in tissue and organ development, differentiation, and function [95].

The advantage of proteomics and metabolomics in anatomic sciences is based on the possibility to provide a comprehensive picture of the proteins (proteome) present in a particular tissue or cell type, which can help to elucidate their functions [96] or even to generate an atlas for human proteins (e.g., Human Protein Atlas proteinatlas.org) [97,98,99].

Proteomics and metabolomics can be used to identify differences in protein expression between different cell types or tissues, which can provide insight into their molecular and functional diversity [100,101]. Both technologies can be used to identify protein–protein interactions, which can help identify metabolite–metabolite interactions and signaling pathways and other complex regulatory mechanisms. Nevertheless, advances in proteomics and more recently in metabolomics and interactomics, have led to the development of new techniques for analyzing and visualizing protein expression data. For example, researchers are more interested in using these omics to study post-translational modifications of proteins and metabolite interactions, which can have significant effects on their functions. Additionally, there is a growing interest in using proteomics to study the relationship between protein expression and disease, or even microbiome and metabolism, which could have implications for a wide range of anatomical studies, leading to the identification of new therapeutic targets.

### 2.5. Radiomics and Novel Imaging Technologies

Radiomics is a rapidly growing field of medical imaging that involves the extraction and analysis of quantitative features from medical images, such as X-ray computed tomography (CT) or magnetic resonance imaging (MRI) scans [43]. These features can be used to characterize the underlying tissue and provide additional information about disease processes, treatment response, and prognosis.

In anatomic studies, radiomics can be used to analyze the relationship between the underlying anatomy and the corresponding radiographic features. For example, radiomics can be used to identify specific features in CT or MRI scans that are associated with the presence of certain anatomical structures or landmarks [26,102].

Radiomics can also be used to study the relationships between different anatomical structures and their corresponding radiographic features [103]. For example, radiomics can be used to identify patterns in CT or MRI scans that are associated with specific types of tumors, or to identify changes in the shape or size of anatomical structures that are associated with certain diseases or conditions.

Overall, radiomics provides a powerful tool for analyzing and understanding the complex relationships between anatomy and radiographic features. By providing quantitative measures of tissue characteristics, radiomics can help clinicians better diagnose and treat disease, and can also facilitate the development of new imaging biomarkers for disease detection and monitoring. Finally, for all the aforementioned reasons, radiomics could become a powerful tool also for medical student education, adding 3D images and organs and systems’ simulations to current education strategies.

Novel imaging technologies have also improved anatomic studies, providing high-resolution and non-invasive imaging of biological specimens. These techniques include X-ray CT, MRI, optical coherence tomography (OCT), and confocal microscopy [104,105,106].

X-ray CT uses X-rays to generate three-dimensional images of tissues and organs, providing detailed information on their internal structure and density. In anatomic studies, X-ray CT can be used to investigate the morphology of bones, teeth, and soft tissues.

MRI uses strong magnetic fields and radio waves to generate high-resolution images of tissues and organs, providing information on their anatomy and functions.

The advantages of novel imaging technologies comprise (i) high-resolution, allowing researchers to visualize structures and tissues at different scales, from molecules to organs; (ii) non-invasiveness, without the need for dissection of tissues and biological samples, particularly useful in case of rare or valuable specimens; (iii) multi-modality with the possibility to combine multiple imaging modalities to provide a more comprehensive understanding of biological structures and tissues; and (iv) time-lapsing, such as confocal microscopy, that allows for capturing the dynamic behavior of biological processes over time.

These technologies are being increasingly used in medical education to create immersive experiences for learning anatomy. They allow students and medical professionals to explore the human body in virtual environments.

The costs of these techniques can be very expensive in terms of purchasing and maintenance of the equipment, expertise and be time consuming for users, resulting in them often being inaccessible to some researchers or institutions. Moreover, the complexity of specific techniques may require specialized knowledge and training to use and interpret the resulting images, limiting their use by non-specialists. Above all this, there are limitations already known by anatomists and microscopists, residing in the limited field of view, as well as for confocal microscopy, where large specimens may need to be imaged in sections, leading to potential artifacts and loss of information. The issues about artifacts and noise can strongly limit the accuracy and interpretability of the resulting images. Finally, especially for radiomics, the use of ionizing radiation can have potential health risks, particularly when imaging human subjects.

In summary, novel imaging technologies have greatly expanded the possibilities for anatomic studies, providing high-resolution and non-invasive imaging of biological specimens. However, their use comes with some limitations, such as cost, complexity, limited field of view, and potential artefacts and noise. Therefore, careful consideration of these pros and cons is essential in deciding the most appropriate imaging technology for a given research question.

### 2.6. Spatial Profiling and Pathomics

Spatial transcriptomics and spatial profiling/multiplex imaging are technologies based on combining different equipment with different objectives, resolutions, costs, and applicability. From the imaging point of view, these techniques enable the study of gene/protein expression patterns within the context of tissue architecture, allowing to investigate of various anatomical structures, such as organs, tissues, and cells. Among all the above-described omics, probably spatial profiling is the approach with the greatest applicability in the anatomical field.

Starting from the results and the approaches from other omics and combining different techniques, such as single-molecule Fluorescence In Situ Hybridization (smFISH), it has been possible to implement the visualization of single molecules within a cell by sequential barcoding through multiple cycles [107]. These procedures have been recently improved by the combination, addition or refinement of other approaches such as sequential FISH (seqFISH) [108], multiplexed error-robust FISH (merFISH) [109], spatial organization of single molecule (osmFISH) [110], and spatially resolved transcript amplicon readout mapping (STARmap) [111].

The commercial availability of several dedicated instruments allowed to diversify and personalize the targets and outcomes of these technologies, from the tagging/barcoding and quantification of oligonucleotides, to the usage of mass cytometry and/or mass spectrometry [112,113,114,115]. In summary, the development and further evolution of spatial molecular profiling technologies, together with new analysis methods, represents a major breakthrough in the field of anatomic studies (among others) and will largely improve biomedical research.

In anatomic studies, spatial transcriptomics can be used to generate gene expression patterns with high-resolution maps across the entire tissue section, which can provide insights into the molecular mechanisms underlying tissue development, homeostasis, and disease. By combining spatial transcriptomics with other imaging techniques, such as microscopy and histology, researchers can gain a more comprehensive understanding of the different anatomical structures and functional organization [116,117,118,119].

For example, spatial transcriptomics has been used to study the molecular architecture of the brain [120], the spatial distribution of immune cells within tumors [121], and the gene expression patterns associated with specific cell types within the gut [122]. These studies have provided valuable insights into the complex biological processes that govern tissue function and have opened up new avenues for the development of targeted therapies for various diseases [123,124,125,126].

The advent of these technique paved the way for Next-Generation Morphometry (NGM) and pathomics that, complementarily with molecular omics techniques, provide tissue-based information on histological structures and integrate multi-omic and high-throughput technologies, computational analysis, and data visualization in the field of anatomical pathology [47]. These approaches, at the early step of their development, provide new insights into disease mechanisms, diagnosis, prognosis, and treatment. From the point of view of anatomical studies, it will become very soon an enormous source of data on histology at an unprecedented scale, generating novel morphology-based research.

### 2.7. Cadaveric Dissections and Preservation, 3D Models and Bioprinting

For preserving cadavers for surgical training and research, several different tools and techniques have been used to slow down decomposition and maintain the anatomical structures for a prolonged period: from the injection of the embalming fluids, typically formaldehyde, into the circulatory system (embalming), to the replacement of bodily fluids and fats with reactive polymers (plastination), and from the formalin fixation to the fresh tissue preservation under controlled conditions and temperature (such as freezing or the usage of commercial preservation solutions). The available resources, and the ethical considerations surrounding the use of human remains for medical education and research together with the intended use of preserved cadavers influenced technological advances in this field, leading to innovative approaches that enhance the quality, longevity, and utility of preserved specimens [127]. To better mimic the natural properties of tissues and organs, novel solutions have been developed to maintain the structural integrity and functional characteristics of preserved specimens over longer periods [128,129,130,131]. Moreover, to monitor and alert users to quality of the preservation, the usage of biosensors can provide real-time information about any changes that might affect the cadavers’/tissues’ conditions, such as pH, temperature, and moisture levels [132,133]. These new tools took advantage not only of the modern techniques on tissue/organ transplantation and of tissue engineering and regenerative medicine, but also novel technologies that are and will soon be able to support and implement this field of anatomical studies. Among them, 3D printing and bioprinting are allowing for the generation of anatomically accurate models and structures using a variety of materials. The use of living cells to create tissue-like structures could eventually lead to the creation of functional tissues for surgical training and research [134]. Nanotechnological approaches to cadaveric preservation are enabling precise control over tissue preservation at the cellular and molecular levels [135,136,137,138]. Nanoparticles and nanomaterials are revolutionizing and enhancing preservation solutions, potentially slowing down tissue degradation and extending their viability [139,140]. More recently, the advent of high-resolution MRI and CT imaging provided detailed 3D reconstructions of anatomical structures, generating accurate virtual reality models for an interactive exploration of anatomical structures and even virtual surgical simulations.

Finally, digital platforms and databases are facilitating the sharing of anatomical data, 3D models, and cadaver dissection, enabling medical professionals worldwide to access valuable resources for cadaver dissections and anatomy education. In line with this data sharing, collaboration platforms, and more recently artificial intelligence algorithms can analyse large datasets of anatomical information, providing insights into optimal techniques and approaches. AI-powered simulations can offer personalized training experiences based on individual skill levels and learning needs.

The combination of these different preservation techniques and tools, such as plastination and 3D printing, or such as the use of biosensors in embalmed or cryopreserved cadavers, can lead to hybrid models that implement the benefits of each approach. These models can have realistic textures, colours, and structural accuracy while being more durable and stable.

### 2.8. Connectomics and Neural Networks

Connectomics focuses on mapping and understanding the connectivity of neural circuits in the brain or other parts of the nervous system, to generate comprehensive and detailed maps of the neural connections between individual neurons, groups of neurons, and brain regions. Connectomics seeks to reveal how information flows and is processed within these networks, contributing to our understanding of brain function and behaviour [141,142]. This omic represents a combination of modern neuroanatomical techniques, such as immunohistochemistry, tracing techniques, Diffusion Tensor Imaging (DTI), Electroencephalography (EEG), Positron Emission Tomography (PET), Magnetoencephalography (MEG), electron/confocal/two-photon microscopy, and computational analysis/neuroinformatic, providing insights into the physical wiring of neural circuits, including the paths of axons, dendrites, and synapses [143].

Connectomics and anatomy are strictly interconnected, since the foundational understanding of the brain’s physical structure is related with the intricate network of connections that underlie brain function [144,145,146]. The synergy between these two fields enhances our comprehension of how neural circuits are organized, how information is transmitted between brain regions, and how these connections contribute to brain functions and behaviour. Detailed knowledge of brain anatomy is essential for interpreting connectomics data. These data help researchers identify the specific regions being studied and the neural pathways involved in information processing. For example, anatomical tracing techniques, such as anterograde and retrograde markers, involve the injection of tracers into specific brain regions to label neural pathways, providing information about the physical routes through which neural signals travel and enhancing our understanding of the anatomical basis of functional connectivity. Moreover, the emergence of microscale connectomics based on electron microscopy (EM) revealed ultrastructural details of synapses and axon–dendrite interactions, providing data about specific wiring patterns and synapse distributions within neural circuits.

Connectomics and neural networks are related concepts with a substantial difference. While the first one is a field within neuroscience that explores the structural connectivity of neural circuits in biological systems, neural networks, in the context of AI, are computational models inspired by the structure of biological neural networks and are used for data analysis, pattern recognition, and prediction [147,148,149,150,151]. While both concepts involve networks of interconnected elements, they operate in different domains and have distinct objectives.

Neural networks consist of input layers, hidden layers (optional), and output layers of interconnected nodes, which are used for solving complex problems by learning patterns and relationships in data through a process called training. Neural networks and anatomy are intertwined at different levels and their interconnections are expanding. For example, the anatomical organization of the brain’s neural networks, including the arrangement of different brain regions, pathways, and synaptic connections, determines how information is processed and transmitted. The architecture of artificial neural networks (ANNs), computational models inspired by the structure and function of biological neural networks, including the arrangement of layers, the number of neurons, and the strength of connections (weights), mimics the way information flows through biological neural networks. Moreover, the function of the network involves mapping the input–output relationships and identifying patterns in the data (Figure 2). This functional mapping reveals how sensory inputs are transformed into motor outputs or cognitive processes [152,153,154].

In the field of neuroembryology and developmental neuroanatomy, researchers use a variety of modern techniques to study the formation and organization of the nervous system during embryonic and foetal development. These techniques allow scientists to investigate the tangled processes that shape the brain and nervous system as they develop over time. Modern techniques such as in utero imaging, transgenic and gene editing, cell labelling and tracing, organoid culture models and neuroimaging of developmental disorders have transformed our understanding of neuroembryology and developmental neuroanatomy, allowing researchers to unravel the intricacies of how the nervous system develops from the earliest stages to its mature form [155,156]. This knowledge has implications for understanding neurodevelopmental disorders, brain plasticity, and potential therapeutic interventions. For example, the combination of the growing amount of histochemical markers and proteins [157,158,159,160,161] such as neurofilament (NFL), microtubule-associated protein (MAP), Tau, synaptic proteins, growth cone (such as GAP-43) and neuronal nuclear markers (such as NeuN, doublecortin and Reelin), and glial fibrillary acidic protein (GFAP), allows pathologists and researchers to assess the health and maturity of neurons, dissect the anatomy of the peripheral nervous system [162] and identify pathological changes associated with various neurological disorders [163].

## 3. Conclusions

The use of omics techniques in anatomic studies offers several advantages over traditional histological and imaging methods. Omics technologies allow the simultaneous analysis of multiple molecular components of tissues and organs, providing a more comprehensive understanding of their structure and function. Moreover, omics techniques are highly sensitive and can detect small changes in molecular composition, making them suitable for detecting early signs of disease or developmental abnormalities. Finally, omics techniques are quantitative, enabling the precise measurement of molecular changes in tissues and organs.

Despite their advantages, omics techniques have several limitations in anatomic studies. One of their main challenges is the complexity of the data generated by omics technologies, requiring specialized computational methods for analysis and interpretation. Moreover, omics techniques may require the destruction of the sample, limiting their use for longitudinal studies or rare specimens. Finally, omics techniques may suffer from technical biases, such as incomplete coverage of the genome or proteome, affecting the accuracy and reproducibility of the results.

Omics techniques and novel technologies have greatly expanded the possibilities for morphological studies, providing researchers with new ways to analyze the molecular and cellular mechanisms underlying biological structures and functions. The advantages of these technologies alone or in combination are (i) the high-throughput analysis, allowing of multiple and/or simultaneous sampling/testing, increasing the efficiency of data collection and analysis; (ii) unbiased analysis, providing a comprehensive analysis of biological structures and functions, which can lead to the identification of new molecular and cellular mechanisms underlying morphological traits; (iii) integration of data, integrating data from multiple sources, such as imaging, molecular profiling, and functional assays, providing a more comprehensive understanding of morphological traits and their underlying mechanisms; and (iv) non-invasiveness, allowing the analysis of biological samples without the need for invasive procedures, which is particularly useful when studying rare or endangered species.

Regarding the use of omics in anatomy, another consideration can be made. Until the last century, anatomical studies benefited from brilliant minds that hypothesized mechanisms and structures, which in some cases were only partially demonstrated or not heeded by the scientific audience until years later. An example in this regard is Cajal’s neuronal hypothesis in contrast to Golgi’s theory that considered the CNS a syncytium. In fact, the debate between these two theories lasted well after the death of both of these two scientists and anatomists, and only the advent of EM allowed Cajal’s perspective to be espoused [164]. In the era of omics, it will undoubtedly be possible to prove or disprove many theories that have only been hypothesized. The downside of this hypertechnologization is that there will be less room for intuitive processes, while interpretive reasoning about the data will be increasingly important.

The limitations of the omics technique can be synthesized into (i) high costs, requiring specialized equipment and personnel, which can limit their accessibility to some researchers and institutions; (ii) technical challenges, needing specific knowledge and technical skills to use and interpret the resulting data, limiting their use by non-specialists; (iii) limited spatial resolution, often providing information at the molecular or cellular level, which may not fully capture the spatial complexity of morphological structures and functions; (iv) data interpretation, requiring sophisticated bioinformatic tools and expertise to analyze and interpret, which can be a bottleneck for some researchers; and (v) sample availability, requiring high-quality biological samples, which may not always be possible, particularly for rare or endangered species.

In summary, omics techniques and novel technologies have the potential to greatly enhance the study of morphology by providing unbiased and comprehensive molecular and cellular analysis of biological structures and functions. Although bearing some limitations, which need to be carefully considered when designing morphological studies, these new technologies may represent an enormous opportunity for the study of anatomy, both macroscopic and microscopic, from the clinical and forensic to the morpho-histological field. A hitchhike to explore new frontiers and scenarios that anatomists cannot miss.

## Figures and Tables

**Figure 1 diagnostics-13-02963-f001:**
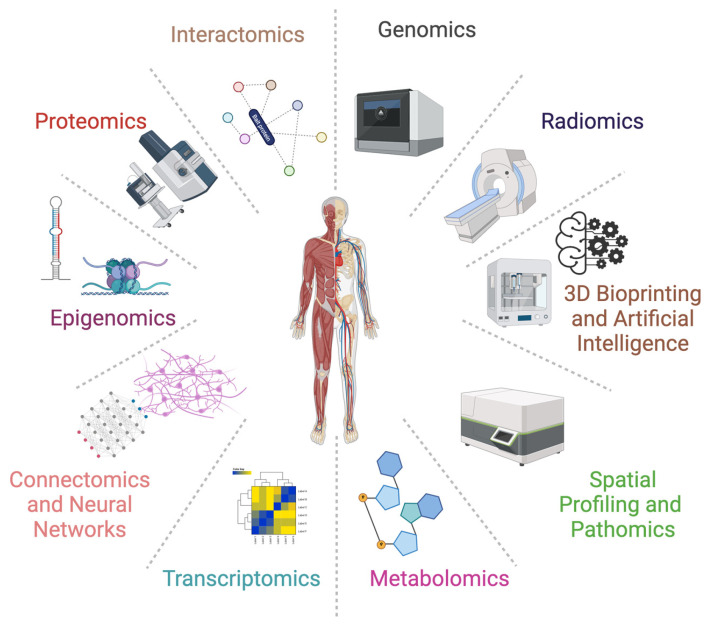
Multi“omic” for anat“omic” studies.

**Figure 2 diagnostics-13-02963-f002:**
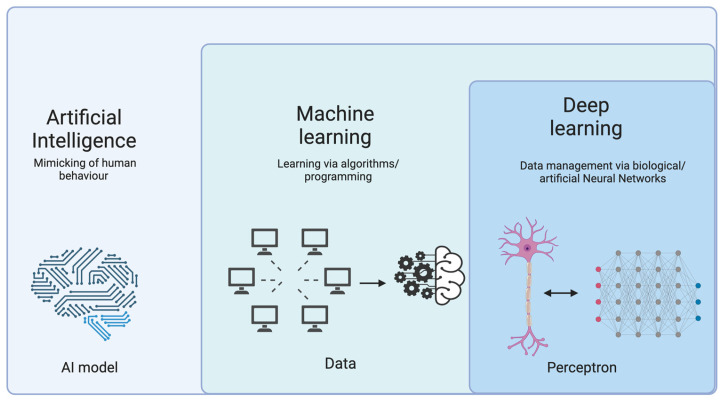
Computer-assisted big data extraction and management.

**Table 1 diagnostics-13-02963-t001:** Omics pros and cons in anatomy.

Omics	Examples of Large-Scale Research Efforts	Utility and Advantages in Science	Utility and Advantages for Anatomical Studies	Major *Caveat*	References
Genomics	1000 Genomes, GWAS * consortia, etc.	Gene-based disease source and direct inference of causality with morbidity/diagnostic/prognosis	Genetic basis of anatomical structures, functions and variations/diversity. Very useful for paleoanthropology.	Still quite expensive and difficult to manage the data. Ethical problems to be considered.	[27,28,29,30,31]
Transcriptomics and Epigenomics	ENCODE * and Roadmap Epigenomics Project, Fantom consortium, MoTrPAC *, COSMIC *	Impact on genotype-phenotype and physiology/patho-physiology and inference of causality	Organ development, differentiation, and function useful for embryology.	Not applicable for all phenotypes	[32,33,34,35]
Proteomics, Metabolomics and Interactomics	CPTAC *, EDRN * and Common Fund	Likely to be very close to the phenotype. Very useful for pathologic studies	Molecular and functional diversity, complex regulatory mechanisms of development; Human Protein Atlas	High costs and difficulty to scale	[36,37,38]
Radiomics and Novel Imaging technologies	AI for health imaging, Enigma consortium	Likely to be very close to the phenotype and measures a combination of genetic and environmental influences. Functional impact and typically easy to infer causality	High-resolution non-invasive more comprehensive understanding of biological structures and tissues	Expensive purchasing of equipment; high complexity	[39,40,41,42,43,44,45,46]
Spatial Profiling and Pathomics	Human Cell Atlas (HCA) consortium	Inexpensive assay for an intermediate step towards the phenotype	Gene expression patterns, high-resolution tissue maps, tissue development, organ/systems homeostasis and structures and functional organization	Very expensive purchasing of equipment; difficult to scale; high complexity	[47,48,49,50,51,52]
Connectomics and neural networks	Human Connectome Project (HCP), Human brain project, MGH/Harvard-UCLA consortium, Human Connectome Project Young Adult	Potential diagnostic, prognostic, and therapeutic interventions in the optic of a personalized and precision medicine	Neurodevelopmental disorders, brain plasticity, and anatomical organization of the brain’s neural networks, including the arrangement of different brain regions, pathways, and synaptic connections	Combination of genetic and environmental influences makes it difficult to infer the direction of causality	[53,54,55,56,57,58,59]

* Acronyms in Table 1: GWAS: Genome Wide Association Studies; ENCODE: Encyclopedia Of DNA Elements; MoTrPAC: Molecular Transducers of Physical Activity Consortium; COSMIC: Catalogue Of Somatic Mutations In Cancer; CPTAC: Clinical Proteomic Tumor Analysis Consortium; EDRN: Early Detection Research Network.

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
