# Peer review of "New Challenges for Anatomists in the Era of Omics"

_diagnostics, 2023, doi:10.3390/diagnostics13182963_

Round 1

Reviewer 1 Report

In this study Stabile and colleagues describe how studies in Anatomy rely upon a wide array of technologies in order to continue to evolve. Although their purpose is laudable, the authors did not achieve success in showcasing the clear co-dependence between anatomy and the different -omics technologies for the progress of anatomy itself and of science in general. The main reason for this is the lack of relevant examples given by the authors in each subsection of the manuscript - the examples given are very broad and do not show in a clear and compelling way how anatomy benefits from the different -omics and vice-versa; in addition the Introduction section is itself poorly written, once again not showing in a compelling way how anatomy has been propelled by technological advances throughout history. Ultimately, this manuscript ends up being more a description of the -omics themselves than a showcase of how they have benefited or will benefit the field of anatomy. In line with this analysis, this reviewer suggests that the authors review the entire manuscript, shorten the description of the -omics technologies and provide more and in depth examples (in the Introduction the mention of other tools/techniques for surgery or cadaveric preserving and the mention of different anatomy schools/anatomists other than italian ones will surely improve the manuscript).

Although the reviewer is not a native English speaker, the quality of English language in this manuscript is good.

Author Response

We thank the reviewer for the suggestions that we wanted to follow.

We emendated the manuscript addressing the suggested examples, enriching with new references in all the different paragraphs. Moreover, we added a table that summarizes the specific advantages of omics in anatomical studies (Table 1).

Moreover, we implemented the introduction section with a summary of the impact of technological advances on anatomical studies throughout history (lines 21-40 and 61-80).

In addition, we added a section with the modern technologies for surgery and cadaveric preservation and named and cited other schools as suggested (lines 312-353).

Reviewer 2 Report

The article "The hitchhiker's guide to the galaxy of omics: new challenges for Anatomists" is a very interesting review about the new challenges of anatomists for technology development and innovations. Indeed this theme is very actual and critical as it is at the center of the necessary evolvement of a very fundamental area as anatomy underling its pivotal role in many different research fields, but also its limitations. The work is well organized and written. It is easy to read and comprehensible.

I would only suggest to the authors to improve the introduction part underlying the recent updates regarding the limitations to the anatomical studies in Italy and other countries (46-55 lines) with some specific and update references. Moreover in this part a brief focus regarding the role of body donation programs and medical education that is strongly connected to scientific research could be added.

With these very limited revisions the manuscript will be even more complete and updated.

Author Response

We thank the Reviewer for positive evaluation of our paper. We added specific details and updated references about limitations and ethical considerations (lines 61-81).

Reviewer 3 Report

The authors attempt to point out in this essay the importance of neuroanatomy in correlation with genetics, imaging and biochemistry in understanding the brain and its disorders. Whereas their purpose is laudable, they ignore some important aspects that also merit discussion. His discussion of anatomy is limited to gross (macroscopical) and microscopical structure, but this is the neuroanatomy of a century ago. There is no mention of modern neuroanatomical techniques of tissue maturation and expression of various molecules in tissue sections and indeed the maturation of individual neurones by immunocytochemistry. In this context, the whole field of neuroembryology and developmental neuroanatomy is barely even mentioned, for example the sequence of synaptogenesis in various regions of the brain and the timing of expression of various proteins in developing neuroblasts its alterations in cerebral malformations (see Sarnat HB. Immunocytochemical markers of neuronal maturation in human diagnostic neuropathology. Cell Tiss Res 2015;359:279-294). There have been many advances in anatomy in the past century and especially the past quarter century that parallel those in molecular genetics and imaging including functional MRI.

Another deficiency is the lack of mention of ultrastructure (transmission and scanning electron microscopy) to understanding subcellular structure in normal and developing cells as well as in pathological conditions. Cajal’s neuronal hypothesis of the 1880’s vs. Golgi’s theory that the CNS is a syncytium was still being debated for decades after the death of both pioneers, until 1956 when Palay confirmed Cajal’s perceptive interpretation, using EM to demonstrate neurones as individual cells with synapses between them and that such connections were not simply artifacts of tissue preparation as seen by light microscopy (Palay SL. Synapses in the central nervous system. J Biophys Biochem Cytol 1956;2(suppl):193-202). The authors’ mention of some historical aspects is admirable, but could be expanded further not only for interest but to contribute to puting the entire theme in perspective.

.

As a neuroscientist, I find the “too cute” title of this article more suitable for a publication for the general public than for a scientific audience. As a result, some readers may not take the article seriously and fail to read beyond its title. A more dignified title would be more appropriate, in my opinion. Similarly, the simple drawing of Fig. 1 is too infantile for a serious scientific publication.

The authors might do well to make a distinction between aetiology and pathogenesis in pathological states. A specific genetic mutation or identification of an infectious agent provide the aetiology of a disease, but the morphogenesis (in the case of developmental disorders) or anatomical alterations (in disorders of mature tissues) define its pathogenesis. Blockage points in metabolic pathways provide more evidence of pathogenesis if the normal function of those pathways is known; the defective gene responsible for the blockage or lack of enzymatic expression is the aetiology.

Notwithstanding my suggestions above, I do agree with most of the text and the points the authors are attempting to express. The references cited are appropriate. The English grammar requires no extensive copy-editing.

Author Response

We thank the reviewer for all the suggestions.

We developed the aspect about modern neuroanatomical techniques partially neglected in our previous version in a subchapter dedicated to Connectomics and Neural Networks. Please find the corrections in the paragraph (lines 355-418).

We followed the suggestion of the reviewer expanding the historical citations and summarizing how technological advanced propelled the development of anatomy, with some perspective considerations. This aspect have been mainly developed in the conclusion section. Please find the corrections in the Conclusion paragraph (lines 448-458).

We modified the title accordingly to the suggestion of the reviewer. The figure 1 has been modified and is a reinterpretation of a famous figure about hallmarks of cancers that has been published on high impact scientific journals. We therefore decided to maintain a simple figure that may summarize the manuscript. In addition, we inserted a new table where we listed the technical aspects discussed in the manuscript. In this way we intend to partially follow the suggestion of the reviewer with a more appropriate scientific language without go far away from the “divulgability” of science, that in our opinion is also extremely important and do not impact on the sobriety of the scientific message(s).

We thank the Reviewer for the accurate suggestions that allowed us to implement and reinforce the message(s) of our manuscript.

Reviewer 4 Report

Omics- technologies bring a lot of new opportunities for anatomists and pathologists. So to be aware about new approaches in genomics, proteomics, metabolomics etc meant to be able to combine these big data to bring the researches into new level. Although the authors tried to cover as much omics- as possible they did nor mention the most important field for anatomists, pathomics, the new era for whole slide images analyses that let the investigators to use microscopes for absolutely another tissue analysis level. I suppose that it requires to add this subchapter into the review to make it more comprehensive.

Author Response

We thank the Reviewer for noticing that we did not explicit the term of pathomics. We integrated the spatial profiling subchapter within the new 4.6 subchapter (Spatial Profiling and Pathomics).

Round 2

Reviewer 3 Report

I thank the authors for their attention to my suggestions and for incorporating most of them into the title and also the text of this manuscript. The article is much improved as a result and is thought-provoking.

Author Response

We thank all the Reviewers for the precious previous critics to the manuscript that allowed us to implement the details and to deepen further aspects of this fascinating new/futuristic world of technologies. This positive evaluation of our revised paper also makes us grateful that this reviewer fully grasped our willingness to provoke new insights about these novel issues and their potential in anatomical studies.